# Men Are from Quartile One, Women Are from? Relative Age Effect in European Soccer and the Influence of Age, Success, and Playing Status

**DOI:** 10.3390/children9111747

**Published:** 2022-11-14

**Authors:** Matthew Andrew, Laura Finnegan, Naomi Datson, James H. Dugdale

**Affiliations:** 1Department of Sport and Exercise Sciences, Institute of Sport, Manchester Metropolitan University, Manchester M15 6BH, UK; 2Football Research Group, Department of Sport and Exercise Science, South East Technological University, X91 HE36 Munster, Ireland; 3Institute of Sport, Nursing and Allied Health, University of Chichester, Chichester PO19 6PE, UK; 4School of Applied Sciences, Edinburgh Napier University, Edinburgh EH14 1DJ, UK

**Keywords:** talent, identification, development, selection, football

## Abstract

The relative age effect (RAE) is characterised by an overrepresentation of athletes born earlier in the selection year. Whilst an RAE is consistently evident in male soccer, examinations in female players remain limited. The aim of the present study was to examine the influence of sex, as well as age, success, and playing status in European soccer players. The sample consisted of a total of 6546 soccer players from 55 soccer nations that competed in recent European Championship qualification campaigns. Results indicated an evident RAE in male [*p* = 0.017] but not female [*p* = 0.765] players. Male players were over-represented by players born in the first quartile for the U17 [*p* < 0.001] and U19 [*p* = 0.001] levels, however, this over-representation did not transfer to senior levels. No RAE was observed at any level for female players. Inside each age group, a slight selection bias towards those born in the first quartile for successful squads was observed but did not significantly differentiate between qualification status for either male or female players. Results from this study highlight the disparity in RAE prevalence between male and female players and raise further questions regarding the value of selecting relatively older players to metrics of success, transition, and selection for senior international soccer.

## 1. Introduction

Soccer is one of the most popular sports worldwide, particularly in Europe. Recently, female participation and popularity have increased exponentially. Between 2016–2017, the number of registered female youth players within Europe increased by ~130,000 (14%; [1]. Female soccer has also experienced an increase in both attendance figures (live and television) and sponsorship, as demonstrated by the creation of relatively new professional leagues (e.g., The FA Women’s Super League in England), resulting in a rise of professionalism in the sport [1]. Accordingly, approaches to talent identification in female soccer may have evolved, but research examining these approaches is limited [2]. Resultant of this exponential growth, approaches to talent identification and causality have been incorrectly extrapolated to the female game, which may be erroneous [3]. Though a continuous growth in research attention on female soccer has been seen, the numbers are not equal to outputs in male soccer. Thus, calls for a growth of research within female soccer; particularly talent identification, have been made (e.g., [2,4]).

For male soccer, a contentious issue related to talent identification is the relative age effect (RAE; [5]). The RAE is characterised by an overrepresentation of players born earlier in the selection year [5]. Researchers suggest that chronologically older athletes may be selected for talent development programmes due to acute anthropometrical and physical advantages or perception of increased skill [6]. However, these advantages often dissipate or reverse at the senior level [7], raising awareness of this process relative to long-term development and success. In contrast, negative implications for relatively younger athletes have been identified, such as higher drop-out rates and limited selection opportunities [8,9]. While this phenomenon has been extensively examined in male players over the last few decades with consistent results (e.g., [10]), examinations in female soccer have been less consistent (e.g., [11]). For example, there was no RAE observed for female US Olympic development program players [12], Division 1 Féminine (France) players [8], or Swiss national team players [13]. Yet others report a significant RAE in female youth international players from Europe and North and Central America [14] and domestic players in Spain [15]. Recently, Götze and Hoppe [16] compared the prevalence of a RAE bias in 1763 male and female national and domestic level players in Germany. Results indicated that an RAE was only prevalent in senior female players competing in the second tier, compared to significant RAE in both male first and second tiers and at youth international level. The inconsistency in observed RAE prevalence, combined with disparities in male samples, raise questions around the pervasiveness of the RAE as an issue related to talent identification and selection in female soccer.

Though Götze and Hoppe [16] directly compared male and female soccer players, they, and other previous reports, focus on a single soccer nation. Most published reports explore RAE prevalence and draw conclusions related to talent identification and development practices in established soccer nations. Acknowledging the substantial differences in talent pools, participation rates, domestic competition, and financial resources it is possible that the RAE prevalence may differ across soccer nations [17]. Typically, reports from established soccer nations report RAE bias at youth levels [16,18,19] but not at senior levels [7] and suggest that RAE prevalence may influence the competitive success [20]. For example, a significant correlation was identified between RAE and success (final league position) in U17 soccer players in Germany [21]. In contrast, only a select number of studies have examined prevalence of an RAE in less established or developing soccer nations. Finnegan et al. [22] examined the birth dates of 1936 U14 male players from Ireland that had been selected for the ‘emerging talent programme’. The authors observed that 68% of selected players were born in the first half of the selection year. Moreover, Dugdale et al. [23] examined birthdate distributions of male soccer players across varying age groups and performance levels in Scotland. These authors observed a significant RAE for players within academy structures but not at the senior professional level. Unsurprisingly, given the paucity of published reports investigating RAE prevalence in female compared to male players, limited data are available concerning RAE for female soccer players across diverse soccer nations.

Lastly, one factor that may moderate RAE prevalence, yet has received little attention, is players that are deemed to possess the skills to compete at higher age levels, also referred to as ‘playing up’. Many studies have independently examined RAE in youth-level soccer and investigations into ‘playing up’ (e.g., [24]); however, research merging these two topics of interest remains scarce. It may be suggested that players that are moved up a chronological age grouping to account for biases typically exacerbated by RAE prevalence and allow for a more appropriate ‘challenge point’ for players at both ends of the selection spectrum (i.e., Q1 and Q4). Subjective evidence posits that athletes that ‘play up’ engage in practice and competition that has a higher yet appropriate challenge point and can have important implications for their developmental outcomes [25]. Whilst theoretically sensical, limited data examine this phenomenon across comprehensive samples of male and female soccer players [24,26].

Accordingly, the aim of the present study was to examine the prevalence of RAE in international female European soccer players. To provide a comparison, we also examined RAE in equivalent international male European soccer players. It was hypothesised that a significant RAE bias would be observed in European male, but not female, players. Furthermore, we hypothesised that an RAE would be prevalent in younger (i.e., U17; U19) but not senior male players, comparable to previous reports [7,23]. Acknowledging the inconsistency in female RAE prevalence, we hypothesised that this observation would not translate to female international players. Given the limited research examining the influence of competition level and ‘playing up’ on RAE in soccer, we forgo making a priori hypotheses.

## 2. Materials and Methods

### 2.1. Participants

Birthdates of 6546 active European soccer players were obtained in March 2021 from the official data centres of the Union of European Football Association [27] and individual nations. Birthdates were collected from all 55 associations under the UEFA governing body and from the qualification squads/teams for the most recent European Championship campaign, respectively (2019 women’s U17 Championships, Bulgaria; 2019 men’s U17 Championships, Ireland; 2019 women’s U19 Championships, Scotland; 2019 men’s U19 Championships, Armenia; 2022 women’s senior Championships, England; 2020 men’s senior Championships, Europe). Players were categorised by sex: female (2387); male (4159), competition level (i.e., age group): U17 (Female = 324; Male = 1187); U19 (Female = 293; Male = 1229); senior (Female = 1770; Male = 1743), qualification status (i.e., did they qualify for the knockout round(s) of their respective competitions): qualified (Female = 994; Male = 1405); non-qualified (Female = 1393; Male = 2754), and by playing status (i.e., they are playing inside, or above their chronological age group): age group (Female = 337; Male = 2020); playing up (Female = 280; Male = 396). Any players that were listed twice (e.g., making appearances at U17 and U19) were categorised based on the most appearances. Because data were freely available via the internet, no approval by an ethical committee was required. The study was conducted in accordance with the declaration of Helsinki. For a full breakdown of the categories and definitions, please see Table 1.

### 2.2. Procedure

The birth month for each player was used to define the birth quarter- and half-year distribution per semester [5]. We adopted cut-off dates defined as: Q1 = Jan-Mar; Q2 = Apr-Jun; Q3 = Jul-Sep; Q4 = Oct-Dec, and semesters: S1 = Jan-Jun; S2 = Jul-Dec. Due to England having different selection cut-off dates to other European countries, we adjusted their quartiles/semesters, specifically (Q1 = Sep-Nov; Q2 = Dec-Feb; Q3 = Mar-May; Q4 = Jun-Aug, and semesters: S1 = Sep-Feb; S2 = Mar-Aug). A failure to be aware of this difference can lead to skewed results within large-scale RAE studies [28].

### 2.3. Data Analysis

The Chi-squared (*χ^2^*) test was used to assess differences between observed and expected birthdate distributions across quartiles for: (1) Each sex irrespective of age or playing level; (2) Each sex, age group, and qualification status; and (3) Each sex and playing status. Expected birthdates were obtained from a European database [29]. They reflected the average population birthdate distributions for all available nations under the UEFA governing body from 1978–2004, capturing the population records for the birth years of the oldest to youngest players within the sample. Population birthdate distributions were identified as: Q1 = 24.4%; Q2 = 25.4%; Q3 = 26.0%; Q4 = 24.2%. Odds ratios (ORs) and 95% confidence intervals (95% CI) were calculated to compare the odds of the frequency of a quartile/semester to another with a reference group consisting of the relatively youngest players (Q4 or S2, respectively). An OR of 1.0 indicated that the frequency is equal in both quarters/semesters whilst an OR of 2.0 indicated that the frequency of one quarter/semester is twice as high as the other [11,16]. ORs were considered significant if the 95% CI range did not include a value <1.00. Furthermore, effect sizes (Cohen’s W) were calculated to determine the magnitude of chi-squared tests, Cohen [30] proposed that where *w* = 0.10, *w* = 0.30, and *w* = 0.50, they specified small, medium, and large effect sizes, respectively. Where appropriate, alpha was set at *p* < 0.05. Data were analysed via SPSS Statistics Version 26.0 for Windows (IBM, Chicago, Illinois, United States).

## 3. Results

There was a statistically significant RAE for male players [*χ^2^* = (*n* = 4159) = 10.2, *p* = 0.017], but not female players [*χ^2^* = (*n* = 2387) = 1.2, *p* = 0.765]. Male players born in the first quartile were over-represented (Q1 vs. Q4, *OR* = 2.4, *CI* = 0.8–3.7 Figure 1), and the ORs declined marginally for comparisons later in the year, with Q4 being inferior for each case (Q2 vs. Q4, OR = 1.7; Q3 vs. Q4, OR = 1.4).

The frequency and percentage distributions of players’ birth quartiles for competition level and qualification status are presented in Table 2. For competition level, irrespective of qualification status, there were statistically significant RAEs for male [U17s: *χ^2^* = (*n* = 1187) = 24.6, *p* < 0.001; U19s: *χ^2^* = (*n* = 1229) = 15.7, *p* = 0.001; Senior: *χ^2^* = (*n* = 1743) = 2.3, *p* = 0.506], but not female players [U17s: *χ^2^* = (*n* = 324) = 1.5, *p* = 0.680; U19s: *χ^2^* = (*n* = 293) = 4.8, *p* = 0.187; Senior: *χ^2^* = (*n* = 1101) = 1.1, *p* = 0.788]. In male players, the chi-squared test indicated significant deviations in birth quartiles for the U17 and U19 levels only, with players born in the first quartile (U17 = 43.2%; U19 = 39.6%) being over-represented (Q1 vs. Q4, U17: *OR* = 4.3, *CI* = 1.8–10.4; U19: *OR* = 2.6, *CI* = 1.2–5.9, Table 2), and the ORs decreased for comparisons later in the year, with Q4 (U17 = 10.0%; U19 = 15.1%) being inferior for each case (Q2 vs. Q4, U17: *OR* = 2.7, *CI* = 1.1–6.8; U19: *OR* = 1.8, *CI* = 0.8–4.2, Q3 vs. Q4, U17: *OR* = 1.9, *CI* = 0.8–5.0; U19: *OR* = 1.2, *CI* = 0.5–2.9). Analysis revealed that although Q1 were over-represented, a significant RAE did not exist for senior players. Further analysis on male U17 and U19 players indicated significant deviations in birth quartiles for both players that had qualified [U17s: *χ^2^* = (*n* = 323) = 32.9, *p* < 0.001; U19s: *χ^2^* = (*n* = 159) = 19.3, *p* < 0.001] and did not qualify [U17s: *χ^2^* = (*n* = 864) = 21.9, *p* < 0.001; U19s: *χ^2^* = (*n* = 1070) = 15.3, *p* = 0.002] for their respective competitions. For players that had qualified, a progressive decrease in birth quartiles from Q1 to Q4 was observed (Q1 vs. Q4, U17: *OR* = 6.1; U19: *OR* = 3.5, Q2 vs. Q4, U17: *OR* = 3.9; U19: *OR* = 2.2, Q3 vs. Q4, U17: *OR* = 2.4; U19: *OR* = 1.7, Table 2), with players that did not qualify following the same pattern (Q1 vs. Q4, U17: *OR* = 3.9; U19: *OR* = 2.5, Q2 vs. Q4, U17: *OR* = 2.4; U19: *OR* = 1.8, Q3 vs. Q4, U17: *OR* = 1.8; U19: *OR* = 1.1, Table 2).

The frequency and percentage distributions of players’ birth quartiles for playing status are presented in Table 3. The chi-squared test indicated significant deviations in birth quartiles for male players playing inside their age group [*χ^2^* = (*n* = 2020) = 20.6, *p* < 0.001], or ‘playing up’ [*χ^2^* = (*n* = 396) = 15.8, *p* = 0.001]. Players born in the first quartile were over-represented (Q1 vs. Q4, Age-Group: *OR* = 3.3; Playing Up: *OR* = 3.2), and the ORs decreased marginally for comparisons later in the year, with Q4 being inferior for each case (Q2 vs. Q4, Age-Group: *OR* = 2.1; Playing up: *OR* = 2.6, Q3 vs. Q4, Age-Group: *OR* = 3.3; Playing up: *OR* = 3.2, Table 3). Analysis revealed that for female players, although there was a greater representation of players in Q1 for playing inside the age groups (Q1 vs. Q4, *OR* = 1.7), this was not significant [*χ^2^* = (*n* = 337) = 4.5, *p* = 0.212]. Moreover, RAE did not exist for players ‘playing up’ [*χ^2^* = (*n* = 280) = 1.0, *p* = 0.806].

## 4. Discussion

The present study explores RAE prevalence in a comprehensive, multi-national sample of both males and females, considering the implications of sex, age, success, and playing status. Our main findings were: (1) Almost no prevalent RAE was observed for female players within our sample; (2) a typical RAE prevalence was observed for equivalent male players within our study, demonstrating a strong RAE bias for youth players and diminishing at senior level; (3) RAE prevalence did not distinguish between qualification status for either male or female players within our sample; (4) RAE prevalence did not distinguish between those who were competing at a higher chronological age level (i.e., ‘playing up’) for either male or female players within our sample.

Analysis of our entire sample (all presently active European female and male soccer players, representing their UEFA nation at all ages) showed a relatively equal birthdate distribution in female soccer players (Figure 1). This finding is only partially consistent with previous literature [8,12,14,15]. Published reports suggest that a lack of RAE prevalence in female soccer could, historically, be due to the reduced popularity of the sport, compared to male soccer, resulting in a smaller talent pool and potentially lower competition for team places [16]. For example, Korgaoker et al. [31] postulated that the strong and consistent RAE in youth female soccer in the United States was due to the popularity of the sport and competition for places within that specific nation. Furthermore, Lagestad et al., [32] found that RAE prevalence increased as females progressed from local to regional teams in Norway. As the popularity of female soccer continues to grow exponentially, we urge those primarily responsible for the identification, development and (de)selection of players, such as coaches and scouts, should be cognisant of this bias as the popularity and participation of female soccer continues to grow. They may look towards other potential predictors of future expert performance, such as physical, skill, psychological and sociological [2], rather than using processes that are evident in the male game and extrapolating them to the female game, which may be erroneous [3].

Other possible explanations for the sex-specific differences in RAE prevalence may be related to the interaction of chronological age, development, and training age [12,33]. For example, female players born in the first half of the year may be more likely to begin playing soccer earlier than their younger counterparts [14], with parents often more hesitant to register a later-born female player into soccer [34]. Although different constructs [35], a ‘maturation-selection’ explanation is often suggested regarding RAE prevalence. [33,35]. Delorme et al., [8] suggests advancing physical development may act as a socially constructed disadvantage for young females, which could lead to dropping out due to feelings of shyness regarding body changes or social pressures to conform to socially constructed sex roles of stereotyped femininity [12]. Moreover, the observed lack of RAE across all ages of competition for female players suggests that talented youth female soccer players are identified through other known predictors of adult high performance in soccer, such as sociological (e.g., hours of practice) or skill (e.g., technical skill) predictors [2]. We propose that additional factors may influence RAE prevalence in female soccer players and encourage researchers and practitioners to be mindful of these considerations when working with this population. 

To provide a comparison, we also examined RAE prevalence in equivalent international male European soccer players. Analysis of the entire male sample showed that players that were born in the first quartile of their selection year were over-represented (Figure 1). This finding is consistent with previous reports of male soccer players across various European domestic leagues and national teams [16,18,19]. We also posited that selection bias would be evident in younger (i.e., U17) male soccer players, yet the advantages of being born earlier in the selection year would not translate to senior levels. In line with this hypothesis, our data indicate an unequal birthdate distribution for U17 male soccer players. This unequal distribution continued until the senior level where it was no longer evident. The strong RAE in male soccer players competing at U17 and U19 competition levels suggests that a selection bias continues to favour chronologically older players despite weakening maturation advantages and disparities in accumulated practice hours. This demonstrates a ‘cascading’ effect of RAE prevalence and suggests a continuation of bias, favouring chronologically older players due to talent identification and (de)selection processes earlier in development [23,35]. The lack of RAE observed in senior male players as they transition from youth to senior level is consistent with previous observations of soccer nations such as Germany [7]. When considering the selection bias at the youth level (e.g., U17, U19), this reduction in RAE prevalence at senior levels challenges the efficacy of this (un)conscious bias and raises awareness of this process relative to long-term development and success [7,36,37].

Analysis of qualification status showed an over-representation of players born at the beginning of the year for squads qualifying for men’s U17 and U19 competitions [16,17,19,21]. Importantly, there were also differences in RAE prevalence for squads that did not qualify for the same competitions. This finding is consistent with previous reports from self-defined smaller and less-established soccer nations [22,23]. It could be suggested that despite differences in population size, participation rate, domestic competition level, and financial resources [17], similar approaches to talent identification and (de)selection might be being adopted [2], demonstrating the pervasiveness of RAE in youth male soccer.

Furthermore, we observed a prominent RAE in male players competing at their chronological age group and those who were ‘playing up’. Recently, Kelly et al. [26] examined factors differentiating youth academy players in England from foundation (U9–U11) or youth (U12–U16) development phases that were either competing at or above their chronological age. Differences were reported for technical and tactical skills for both phases, as well as differences in physical and psychological characteristics within the youth development phase. However, it was noted that ~80% of players that were ‘playing up’ were born in the first half of their initial selection year. In the youth development phase in England, enhanced maturity status (greater percentage of estimated adult height attained) is suggested to contribute to ‘playing up’ [26]. Within youth sports, it is possible to have two players with the same relative age yet have a wide variance in biological age [38]. Early maturing players in youth soccer have been found to be taller and heavier than late-maturing players [39]. Relatively high numbers of early-maturing athletes within chronologically younger athlete samples have been identified [40]. This suggests that the relatively younger players who exhibit advanced growth and maturation have an increased likelihood of selection into soccer development pathways [41], perhaps nullifying the negative impact of a later birthdate for early maturing players. Finally, Kelly et al. [26] suggest that when ‘playing up’, players were at least a full chronological year younger than their peers which may create an ‘underdog’ effect. This may result in relatively young athletes engaging in higher levels of practice and play to match/surmount greater technical/tactical or psychological performance of their older peers [24]. Consequently, it has been reported that younger athletes who progress to being professional have been awarded more accolades and have longer-lasting careers compared to their older peers [42]. We propose that multidimensional factors associated with RAE prevalence may influence both initial selection and decisions pertaining to ‘playing up’ in male youth soccer.

This study is not without its limitations. First, due to obtaining birth data from external sources, we did not obtain any anthropometric data. Thus, our supposition of chronologically older players possessing greater physical, cognitive, and psychosocial attributes and other skills was based on previous theoretical assumptions rather than original data. Though some researchers have questioned whether it is conducive to continue to examine the relative age effect [43], findings from the present study provide a comprehensive evaluation of female players across multiple nations and an equivalent male comparison. Further, the present study explores the influence of RAE bias on previously identified areas of interest to talent identification and development (i.e., ‘playing up’ and competitive success).

## 5. Conclusions

In summary, our results demonstrate that the RAE exists in male, but not female, soccer players participating in UEFA European tournaments between 2019–2022. Male soccer players competing at U17 and U19 levels were over-represented by players born at the beginning of their selection year. This bias did not transition into senior level, questioning the efficacy of this (un)conscious bias. However, these advantages often dissipate or reverse at the senior level [7]. Furthermore, within these age groups, this selection bias did not discriminate against those that qualified for their respective competitions or whether players were playing at a higher chronological age (i.e., ‘playing up’).

## Figures and Tables

**Figure 1 children-09-01747-f001:**
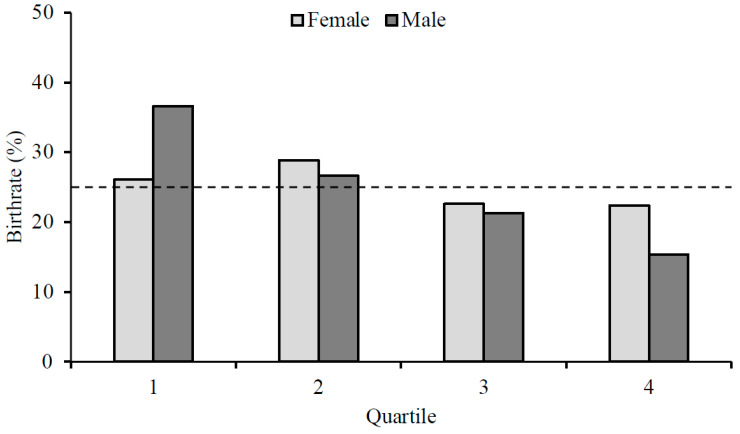
Birth quartile distribution for female (light grey) and male (dark grey) soccer players.

**Table 1 children-09-01747-t001:** Categories, subcategories, and definitions of independent variables.

Category	Subcategory	*n*	Definition
Sex	Female	2387	Represented their respective nation’s female soccer team.
	Male	4159	Represented their respective nation’s male soccer team.
Competition Level	U17	1511	Represented their nation during the U17 qualification campaign.
	U19	1522	Represented their nation during the U19 qualification campaign.
	Senior	2513	Represented their nation during the Senior qualification campaign.
Qualification Status	Qualified	2399	Qualified for the knockout round(s) of their respective competitions.
	Non-Qualified	4147	Did not qualify for the knockout round(s) of their respective competitions.
Playing Status	Age Group	2357	Playing inside their age group (e.g., 16/17 years playing at U17 level).
	Playing Up	676	Playing above their age group (e.g., 16/17 years playing at U19 level).

**Table 2 children-09-01747-t002:** Birth quartile distribution by sex, age group, and qualification status (* Significant at an alpha level of *p* < 0.05).

				Birthdate Distribution (%)				Odds Ratio (95% CI)					
			*n*	Q1	Q2	Q3	Q4	Q1 vs. Q4	Q2 vs. Q4	Q3 vs. Q4	S1 vs. S2	χ^2^	*p*
Female	U17	Qual	165	42 (25.5)	52 (31.5)	37 (22.4)	34 (20.6)	1.2 (0.6–2.8)	1.5 (0.7–3.4)	1.1 (0.5–2.5)	1.3 (0.8–2.3)	2.6	0.467
		Non	159	45 (28.3)	42 (23.3)	37 (23.3)	35 (22.0)	1.3 (0.6–2.8)	1.2 (0.5–2.7)	1.1 (0.5–2.4)	1.2 (0.7–2.1)	1.3	0.735
		All	324	87 (26.9)	94 (29.0)	74 (22.8)	69 (21.3)	1.3 (0.6–2.8)	1.4 (0.6–3.0)	1.1 (0.5–2.4)	1.3 (0.7–2.2)	1.5	0.68
	U19	Qual	160	51 (31.9)	37 (23.1)	37 (23.1)	35 (21.9)	1.5 (0.7–3.2)	1.1 (0.5–2.4)	1.1 (0.5–2.4)	1.2 (0.7–2.1)	3.0	0.383
		Non	133	46 (34.6)	39 (29.3)	26 (19.5)	22 (16.5)	2.1 (0.9–4.7)	1.8 (0.8–4.0)	1.2 (0.5–2.8)	1.8 (1.0–3.1)	8.9 *	0.03
		All	293	97 (33.1)	76 (25.9)	63 (21.5)	57 (19.5)	1.7 (0.8–3.7)	1.3 (0.6–3.0)	1.1 (0.5–2.5)	1.4 (0.8–2.5)	4.8	0.187
	Sen	Qual	669	165 (24.7)	201 (30.0)	144 (21.5)	159 (23.8)	1.1 (0.5–2.3)	1.3 (0.6–2.7)	0.9 (0.4–2.0)	1.2 (0.7–2.1)	1.6	0.654
		Non	1101	275 (25.0)	317 (28.8)	259 (23.5)	250 (22.7)	1.1 (0.5–2.4)	1.3 (0.6–2.8)	1.0 (0.5–2.3)	1.2 (0.7–2.0)	0.8	0.849
		All	1770	440 (24.9)	518 (29.3)	403 (22.8)	409 (23.1)	1.1 (0.5–2.4)	1.3 (0.6–2.8)	1.0 (0.4–2.2)	1.2 (0.7–2.1)	1.1	0.788
Male	U17	Qual	323	147 (45.5)	94 (29.1)	58 (18.0)	24 (7.4)	6.1 (2.4–15.9)	3.9 (1.5–10.4)	2.4 (0.9–6.7)	2.9 (1.6–5.3)	32.9 *	<0.001
		Non	864	366 (42.4)	230 (26.6)	173 (20.0)	95 (11.0)	3.9 (1.6–9.2)	2.4 (1.0–5.9)	1.8 (0.7–4.6)	2.2 (1.3–4.0)	21.9 *	<0.001
		All	1187	513 (43.2)	324 (27.3)	231 (19.5)	119 (10.0)	4.3 (1.8–10.4)	2.7 (1.1–6.8)	1.9 (0.8–5.0)	2.4 (1.3–4.3)	24.6 *	<0.001
	U19	Qual	159	66 (41.5)	41 (25.8)	33 (20.8)	19 (11.9)	3.5 (1.5–8.1)	2.2 (0.9–5.2)	1.7 (0.7–4.3)	2.1 (1.2–3.7)	19.3 *	<0.001
		Non	1070	421 (39.3)	297 (27.8)	186 (17.4)	166 (15.5)	2.5 (1.1–5.7)	1.8 (0.8–4.1)	1.1 (0.5–2.7)	2.0 (1.2–3.6)	15.3 *	0.002
		All	1229	487 (39.6)	338 (27.5)	219 (17.8)	185 (15.1)	2.6 (1.2–5.9)	1.8 (0.8–4.2)	1.2 (0.5–2.9)	2.0 (1.2–3.6)	15.7 *	0.001
	Sen	Qual	923	295 (32.0)	239 (25.9)	219 (23.7)	170 (18.4)	1.7 (0.8–3.9)	1.4 (0.6–3.2)	1.3 (0.6–2.9)	1.4 (0.8–2.4)	4.0	0.265
		Non	820	229 (27.9)	208 (25.4)	215 (26.2)	168 (20.5)	1.4 (0.6–3.0)	1.2 (0.6–2.8)	1.3 (0.6–2.9)	1.1 (0.7–2.0)	1.1	0.784
		All	1743	524 (30.1)	447 (25.6)	434 (24.9)	338 (19.4)	1.6 (0.7–3.4)	1.3 (0.6–3.0)	1.3 (0.6–2.9)	1.3 (0.7–2.2)	2.3	0.506

**Table 3 children-09-01747-t003:** Birth quartile distribution by sex and playing status (* Significant at an alpha level of *p* < 0.05).

				Birthdate Distribution (%)				Odds Ratio (95% CI)					
			*n*	Q1	Q2	Q3	Q4	Q1 vs. Q4	Q2 vs. Q4	Q3 vs. Q4	S1 vs. S2	χ^2^	*p*
Female	Age Group		337	106 (31.5)	96 (28.5)	72 (21.4)	63 (18.7)	1.7 (0.8–3.7)	1.5 (0.7–3.4)	1.1 (0.5–2.6)	1.5 (0.9–2.6)	4.5	0.212
	Playing Up		280	78 (27.9)	74 (26.4)	64 (22.9)	64 (22.9)	1.2 (0.6–2.7)	1.2 (0.5–2.5)	1.0 (0.5–2.2)	1.2 (0.7–2.1)	1.0	0.806
Male	Age Group		2020	851 (42.1)	542 (26.8)	369 (18.3)	258 (12.8)	3.3 (1.4–7.6)	2.1 (0.9–5.0)	1.4 (0.6–3.5)	2.2 (1.3–4.0)	20.6 *	<0.001
	Playing Up		396	149 (37.6)	120 (30.3)	81 (20.5)	46 (11.6)	3.2 (1.4–7.7)	2.6 (1.1–6.3)	1.8 (0.7–4.4)	2.1 (1.2–3.8)	15.8 *	0.001

## Data Availability

The data that support the findings of this study are available on request from the corresponding author. The data are not publicly available due to restrictions (e.g., containing information that could compromise the privacy of research participants).

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
