# Peer review of "Men Are from Quartile One, Women Are from? Relative Age Effect in European Soccer and the Influence of Age, Success, and Playing Status"

_children, 2022, doi:10.3390/children9111747_

Round 1

Reviewer 1 Report

Very good work, well done the number of participants you reached is great. The authors might mention the possible effect of genetic change on  on competitive success. This could open avenues for further study. page 7 306

Author Response

Reviewer 1

Reviewers Comment 1

Very good work, well done the number of participants you reached is great. The authors might mention the possible effect of genetic change on  on competitive success. This could open avenues for further study. page 7 306

Authors Response

Thank you for taking time to consider our manuscript and for the positive response

Reviewer 2

Abstract

Reviewer Comment 1

The results presented in the abstract should be supported by their statistic. 

Authors Response

            We thank the reviewer and have now included all relevant statistical values.

Introduction

Reviewer Comment 2

95-100. This information should be addressed in the Methods section. Otherwise, you are going to be redundant.

Authors Response

We acknowledge that this level of detail is better suited in the methods. We have therefore merged this with the relevant section in (Lines 121-127).

Methods

Reviewer Comment 3

145-149: The authors should provide more information about what type of test used to calculate the odds ratio and confidence intervals to compare the odds of the frequency of a quartile/semester to another reference group.

Authors Response

We thank the reviewer for their comment. We have now included a small statement (supported by references) to indicate how we compared the odds of the frequency of a quartile or semester to another (Lines 154-157).

Reviewer Comment 4

The authors should provide a deeper description of how they create the category “playing status” related to playing up. It would be advisable to create a table where all the independent and dependent variables of the study were described.

Authors Response

We thank the reviewer for this comment, we have now added a small table (Table 1) to support or description of each of the independent and dependent variables.

Reviewer Comment 5

148-150. Please provide a citation that support the use of Cohen´s W in Chi-squared analysis and why the authors use these specific ranges for the effect size. 

Authors Response

Our use of Cohen’s W was based on Cohen (1992) and has been utilized in our previous work (Finnegan et al., 2017), we have now included this reference within the text and reference section.

Results

Reviewer Comment 6

154-158: In addition to the OR, the Confidence interval for the Odds ration should be added.

Authors Response

            We have now included the 95% CI to all relevant results.

Discussion

Reviewer Comment 7

The discussion is well documented and supported by scientific evidence. However, its structure should be reviewed. The authors start talking about female players first, male players second to come back to female players again and finally, to male players. It would be appropriate to focus on female players first, discussion all the findings and then, focus on male players. This structure could be clearer for the readers to understand all the findings.

Authors Response

We thank the reviewer for this comment. Upon reflection and rereading, we can see how this would be easier for the reader. As the focus is on female soccer, we have aimed the first few paragraphs at female soccer players, then focused on male players.

Reviewer Comment 8

I would recommend removing the mentions to tables during the discussion.

Authors Response

            We have removed any references to Tables throughout the discussion.

Reviewer Comment 9

312: “This bias did not transition into senior level questioning the efficacy of this (un)conscious bias”. This sentence is speculative and should be addressed in the discussion section, not in the conclusions. During the discussion, authors should reflect on why this fact should be questionable. 

Authors Response

We thank the reviewer for this comment and agree this needed some expansion, we have therefore included more information around this area (Lines 275-280).

Reviewers Comment 10 

315-318. This part of the conclusion should be added as part of the discussion and particularly in the section of practical applications. Also, It is necessary to expand this section by explaining what this study means for coaches and practitioners.

Authors Response.

We understand and respect the reviewers request here. Given the data collected, as well as the limited literature examining elite youth female soccer player, we believe it would not warrant a full paragraph. This lack of literature and data has been highlighted recently (Emmonds et al., 2019; Williams et al., 2020; Kryger et al., 2021; Randell et al., 2021). Therefore, as an alternative we have expanded on this comment and acknowledged what this would mean by scouts and coaches.

Reviewer 3

Reviewer Comment 1

The theme of the article is not very clear, and the background introduction is not detailed enough. There are many places where the expression needs to be more precise, such as lines 11, 23 and 33. The discussion needs to be further modified and improved in breadth and depth. The author is suggested to make changes.

Authors Response

We thank the reviewer for the comment. We agree and upon reflection, the theme (i.e., relative age in women’s European soccer) is not as clear as it could be. The research team have considered the emphasis of the article and amended accordingly to ensure the theme of female soccer is predominant. We have substantially developed and rearranged the introduction to consider this. In terms of the discussion, we have restructured to improve clarity, breadth, and depth, as per reviewer suggestion. We hope this is suitable to the reviewer, should they wish us to reviewer further, please let us know.

Reviewer 2 Report

The study aimed to examine the prevalence of RAE in female and male soccer players in Europe. The novelty of this paper is to provide more information about this phenomenon in women, so that there is already strong scientific evidence in men players. The study is well conducted and adds interesting conclusions that compare this effect in male and female players.

Abstract:

The results presented in the abstract should be supported by their statistic.

Introduction:

95-100. This information should be addressed in the Methods section. Otherwise, you are going to be redundant.

Methods:

145-149: The authors should provide more information about what type of test used to calculate the odds ratio and confidence intervals to compare the odds of the frequency of a quartile/semester to another reference group.

The authors should provide a deeper description of how they create the category “playing status” related to playing up. It would be advisable to create a table where all the independent and dependent variables of the study were described.

148-150. Please provide a citation that support the use of Cohen´s W in Chi-squared analysis and why the authors use these specific ranges for the effect size.

Results:

154-158: In addition to the OR, the Confidence interval for the Odds ration should be added.

Discussion:

The discussion is well documented and supported by scientific evidence. However, its structure should be reviewed. The authors start talking about female players first, male players second to come back to female players again and finally, to male players. It would be appropriate to focus on female players first, discussion all the findings and then, focus on male players. This structure could be clearer for the readers to understand all the findings.

I would recommend removing the mentions to tables during the discussion.

312: “This bias did not transition into senior level questioning 312 the efficacy of this (un)conscious bias”.

This sentence is speculative and should be addressed in the discussion section, not in the conclusions. During the discussion, authors should reflect on why this fact should be questionable.

315-318. This part of the conclusion should be added as part of the discussion and particularly in the section of practical applications. Also, It is necessary to expand this section by explaining what this study means for coaches and practitioners.

Author Response

(The authors gave the same response as above.)

Reviewer 3 Report

The theme of the article is not very clear, and the background introduction is not detailed enough. There are many places where the expression needs to be more precise, such as lines 11, 23 and 33. The discussion needs to be further modified and improved in breadth and depth. The author is suggested to make changes.

Author Response

(The authors gave the same response as above.)
